# Selective Heterogeneous Fenton Degradation of Formaldehyde Using the Fe-ZSM-5 Catalyst

**DOI:** 10.3390/molecules29122911

**Published:** 2024-06-19

**Authors:** Peiguo Zhou, Jiaxin Hou, Donghui Zhang, Ziqiao Liao, Liping Yang, Wenjing Yang, Xin Ru, Zongbiao Dai

**Affiliations:** College of Ecology and Environment, Nanjing Forestry University, Nanjing 210037, China; houjiaxin990117@163.com (J.H.); 15996313359@163.com (D.Z.); yuki.asang.zq@hotmail.com (Z.L.); kikicurtis_77@yeah.net (L.Y.); yyywj7089@163.com (W.Y.); 13592585561@139.com (X.R.); dzongbiao@163.com (Z.D.)

**Keywords:** Fe-ZSM-5, heterogeneous Fenton, selective, formaldehyde

## Abstract

As a toxic Volatile Organic Pollutant (TVOC), formaldehyde has a toxic effect on microorganisms, consequently inhibiting the biochemical process of formaldehyde wastewater treatment. Therefore, the selective degradation of formaldehyde is of great significance in achieving high-efficiency and low-cost formaldehyde wastewater treatment. This study constructed a heterogeneous Fe-ZSM-5/H_2_O_2_ Fenton system f or the selective degradation of target compounds. By immobilizing Fe^3+^ onto the surface of a ZSM-5 molecular sieve, Fe-ZSM-5 was prepared successfully. XRD, BET and FT-IR spectral studies showed that Fe-ZSM-5 was mainly composed of micropores. The influences of different variables on formaldehyde-selective heterogeneous Fenton degradation performance were studied. The 93.7% formaldehyde degradation and 98.2% selectivity of formaldehyde compared with glucose were demonstrated in the optimized Fenton system after 360 min. Notably, the resultant selective Fenton oxidation system had a wide range of pH suitability, from 3.0 to 10.0. Also, the Fe-ZSM-5 was used in five consecutive cycles without a significant drop in formaldehyde degradation efficiency. The use of reactive oxygen species scavengers indicated that the hydroxyl radical was the primary active species responsible for degrading formaldehyde. Furthermore, great degradation performance was acquired with high concentrations of formaldehyde for this system, and the degradation efficiency was more than 95.0%.

## 1. Introduction

The expansion of industrial activities has adversely impacted water quality and aquatic ecosystems [1]. Volatile Organic Compounds (VOCs), for instance, halocarbons, aldehydes, ketones, and polycyclic aromatic hydrocarbons, are among the most harmful water pollutants [2]. Industrial manufacturing products, such as resin, plastic, paper making, artificial fiber, anti-corrosive leather plywood, etc., will discharge a large amount of formaldehyde wastewater [3,4,5]. Formaldehyde has a strong toxic effect on living organisms. If people drink water contaminated by formaldehyde for a long time, it will cause dizziness, anemia, and various other neurological and mental disorders. Formaldehyde is difficult to biodegrade because it can inhibit the activity of microorganisms. Therefore, there is a critical need to explore efficient purification technologies to treat industrial effluents containing formaldehyde [6].

Currently, formaldehyde wastewater is mainly treated using the Fenton oxidation technique [7], photocatalytic oxidation technique [8], oxidation technique [9], chlorine dioxide oxidation technique [10], and other methods. The Fenton oxidation technique is one of the Advanced Oxidation Processes (AOPs) applied for the degradation of organic pollutants that are difficult to biodegrade in wastewater [4,11,12,13]. The traditional homogeneous Fenton has been widely used in wastewater treatment. However, it has some disadvantages, such as the large consumption of reagents and high operating costs [14,15]. The Fenton oxidation technique can degrade refractory pollutants completely or partially through the decomposition of H_2_O_2_ to produce •OH [16,17,18,19,20,21,22]. Kajitvichyanukul P. et al. [4] established that formaldehyde was almost completely mineralized by photo-Fenton, and the environmental toxicity was at a significantly low level after treatment for 240 min. Moussavi G. et al. [23] showed that the complete degradation of formaldehyde (8000 mg L^−1^) was achieved in heterogeneous Fenton after 6 min.

Furthermore, the hydrolysis of Fe(III) is usually performed under acidic conditions (pH 3.0), so the application of conventional Fenton oxidation techniques is limited to a narrow pH range [24]. Therefore, the development of heterogeneous Fenton catalysts is necessary for heterogeneous Fenton processes based on Fenton chemistry. Fe-based heterogeneous Fenton catalysts are the most widely used catalyst in Fenton technology, which can effectively extend the pH suitability of Fenton application to neutral conditions. For instance, Mohammadifard Z. et al. [25] proposed that combining visible light and H_2_O_2_ with MIL-100(Fe) could degrade 93.0% of formaldehyde (700 mg L^−1^). Macdonald J. et al. [5] prepared the zeolite Y-supported Fe catalyst and used it to degrade formaldehyde by photo-Fenton, which showed that the degradation of formaldehyde was still good in neutral solution. In addition, designing heterogeneous catalysts to achieve highly selective Fenton techniques is of significant practical importance [26,27].

At the same time, achieving highly selective oxidation is a scientific challenge. Playing an oxidizing role in the Fenton reaction, •OH has a high oxidation potential of 2.73 V (E^0^(HO•^+^H^+^/H_2_O)) that can oxidize a wide range of organic substances [28]. As a result, organics other than target contaminants inevitably consume a substantial portion of •OH, and the demand for reagents (hydrogen peroxide and iron catalyst) is large [14,15]. Adhish J. et al. [29] reported an excellent heterogeneous Fenton catalyst, copper sulfide nanoparticles, for the highly selective degradation of dyes. Lei Chen et al. [30] utilized the selective interfacial oxidation of methylene blue and chrysoidine G, which was based on the heterogeneous Fenton catalytic properties of Fe-OCNT nanocomposites.

ZSM-5 was first developed by the American Mobile company in 1972, which is a new zeolite molecular sieve with a high-silicon three-dimensional channel. The zeolite molecular sieve (ZSM) has high thermal and hydrothermal stability, with most of its pore sizes measuring approximately 0.55 nm, classifying it as a medium-pored zeolite. Because of its shape selection, catalytic performance, and unique surface acidity, it is widely used in adsorption separation, cation exchange, catalytic degradation, and so on. The unique pore structure of the ZSM-5 provides the crystal structure basis for the design of highly selective and highly active catalysts. In the field of wastewater treatment, the ZSM-5 with a silicon-to-aluminum ratio of 220 to 400 is commonly used to catalyze the degradation of organic pollutants in wastewater, such as phenol, formaldehyde, and methylene blue trihydrate. Through ion exchange, metal doping, and pore expansion of the ZSM-5 zeolite, the efficient catalytic degradation of organic pollutants in wastewater can be achieved [31]. Xin Xing [32] studied the Cu-ZSM-5 as a selective catalyst for the degradation of n-butylamine. The results showed that one of the n-butylamine degradation products could reach 90.0% at 350 °C, which suggested that the Cu-ZSM-5 catalyst could remove VOCs selectively. Lin-Cong He [33] studied the selective oxidation of diethylamine using the MnO_x_/CeO_2_/ZSM-5 catalyst.

In this study, the selective catalyst Fe-ZSM-5 was prepared by the infiltration method. The Fe-ZSM-5/H_2_O_2_ heterogeneous Fenton system was constructed for the selective oxidation degradation of formaldehyde. The synthesized catalyst was fully characterized using spectroscopic and microscopic techniques, including XRD, FT-IR, BET, and SEM. Based on the results, the Fe-ZSM-5 catalyst had a fine microporous structure and a phase composition identical to that of the ZSM-5. The catalyst still exhibited excellent degradation performance on formaldehyde during multiple reuses and exhibited a low iron leaching rate. Importantly, the heterogeneous Fe-ZSM-5/H_2_O_2_ Fenton system showed a broad pH suitability and high selectivity for formaldehyde. The underlying mechanism of such systems had been inferred. This system provided the possibility for the efficient and low-energy treatment of formaldehyde wastewater, helping to reduce the production of secondary pollutants and the use of chemical reagents, so as to achieve sustainable development by reducing energy consumption and secondary pollution to the environment.

## 2. Results and Discussion

### 2.1. Fe^3+^ Loading Condition of Catalyst

#### 2.1.1. Influence of Different Impregnation Concentration on Catalyst Performance

It can be seen from Figure 1a that the iron load was increased with an increase in the impregnation concentration of iron ion, because the driving force, which was the difference in the iron ion concentration, made the adsorption amount of iron ion on ZSM-5 molecular sieve or the ion exchange capacity on the surface of the ZSM-5 molecular sieve gradually increase. However, when the impregnation concentration of Fe(NO_3_)_3_ solution was increased from 0.8 mol/L to 1.6 mol/L, the iron load of the ZSM-5 molecular sieve did not increase significantly, which could be due to the fact that the adsorption of iron ions on the ZSM-5 molecular sieve tended to be saturated, and the iron load reached a stable state. At this time, increasing the concentration of Fe(NO_3_)_3_ solution had little effect on the iron loading capacity of the catalyst, and the iron loading efficiency was not high. The optimal impregnation concentration was 0.8 mol L^−1^ of Fe(NO_3_)_3_ solution from an economic point of view and the efficiency of carrying iron.

#### 2.1.2. Influence of Different Impregnation Temperatures on Catalyst

As can be seen from Figure 1b, when the temperature rose to 50 °C, the iron loading of the ZSM-5 molecular sieve was increased. Due to the higher solution temperature being increased, the internal energy and the molecular movement became violent, which was conducive to the entry of iron ions into the ZSM-5 molecular sieve and promoted its iron loading process. But, when the temperature continued to rise from 50 °C to 95 °C, the iron load of the ZSM-5 molecular sieve was decreased, and too high a temperature would have a negative effect on the iron load process. Overly high temperatures can cause the adsorbed iron ions to desorb. Therefore, 50 °C was the best water bath temperature for iron loading.

#### 2.1.3. Influence of Different Impregnation Liquid-to-Solid Ratios on Iron Loading

It can be seen from Figure 1c that with an increase in the impregnation liquid-to-solid ratio, the iron load on the ZSM-5 molecular sieve was increased. The iron loading was obviously increased, while the ratio of liquid to solid was increased from 10:1 to 15:1. However, when the ratio of liquid to solid continued to increase, the improvement effect on the iron loading became weak, fluctuating or slightly decreasing. From an economic point of view and considering the iron carrying efficiency, the impregnation liquid–solid ratio of 15:1 was more appropriate.

#### 2.1.4. Effect of Different Impregnation Times on Iron Load

As can be seen from Figure 1d, the iron load on the ZSM-5 molecular sieve was increased with the increase in impregnation time. After calculation, the increase percentage of iron load of 2 h immersion time compared with that of 1 h immersion time was 3.9%, the increase percentage of iron load of 4 h immersion time compared with that of 2 h immersion time was 14.9%, and the increase percentage of iron load of 8 h immersion time compared with that of 4 h immersion time was 13.0%. It can be seen that the iron load efficiency was highest at an immersion time of 4 h. The iron loading efficiency began to decrease gradually with additional time, so the 4 h impregnation time was selected as the optimal time.

### 2.2. Characterization of Catalysts

The shape, size, chemical composition, and crystal structure of the synthesized Fe-ZSM-5 nanoparticles were determined through various evaluation techniques. They are presented in the following paragraphs.

The XRD patterns (Figure 2a) of the Fe-ZSM-5, ZSM-5, and Fe_2_O_3_ nanoparticles (NPs) were acquired using an XRD analyzer (Ultima IV, Rigaku, Japan). The Fe-ZSM-5 characteristic peaks appeared at 2θ = 23.9°, 23.3°, 23.0°, 8.8°, and 7.9°, corresponding to the (303), (051), (332), (200), and (101) crystal planes of the ZSM-5, respectively. The characteristic peaks of iron-rich lepidolite were found in the samples at 2θ = 8.8°, 26.8°, 36.1°, and 45.6°, which were associated with the crystal planes of (001), (003), (113), and (005), respectively. Three typical peaks were observed for Fe_2_O_3_ NPs at 2θ = 24.1°, 33.1° and 35.6°, respectively, which coincided with previous research. The weak peak strength was due to the low iron content or iron only adhering to the surface of the catalyst [12]. The characteristic peaks of Fe-ZSM-5 were consistent with the characteristic peaks of Fe_2_O_3_ NPs and ZSM-5, indicating the successful synthesis of the Fe-ZSM-5 catalyst.

When P/P_0_ < 0.05 (Figure 2c,d), N_2_ adsorption was increased rapidly, indicating that a large number of microporous structures existed within both ZSM-5 and Fe-ZSM-5. The average pore size diameter of Fe-ZSM-5 was approximately 2.247 nm, further proving that this catalyst was mostly composed of micropores (Table 1) [34]. The catalytic performance of the catalyst is positively correlated with the number of active sites. After loading Fe, the pore volume and the specific surface area of micropores were reduced by 0.6% and 1.8%, respectively, indicating that the number of active sites was decreased in comparison to ZSM-5.

The FT-IR spectra (Figure 2b) show that the process of loading iron did not change the internal functional groups of the ZSM-5, and the loading might only occur on the surface of the ZSM-5. The bands at 800 cm^−1^ and 1074 cm^−1^ were attributed to the Si-O-Si stretching modes of Fe-ZSM-5 [35]. The band at 544 cm^−1^ was assigned to the unique chain structure unit of the ZSM-5, which consists of eight five-membered rings connected by edges to form a symmetrical structure [36]. Another band was detected at 431 cm^−1^, which was related to the Si-O bending vibration of the ZSM-5 molecular sieve.

The SEM images (Figure 3) show that the Fe-ZSM-5 exhibits a smooth surface with a regular shape and uniform pore structure, similar to ZSM-5 [35]. Many fine particles can be clearly observed on the surface of the ZSM-5 (Figure 3c), which reveals that Fe was uniformly attached to the grain surface, with no apparent agglomeration. Although a few cracks appear on the surface of the Fe-ZSM-5 catalyst (Figure 3d), the grain shape and structure remain intact after the Fenton reaction.

### 2.3. Degradation of Formaldehyde in the Fe-ZSM-5/H_2_O_2_ System

#### 2.3.1. Effect of Experimental Parameters on Degradation of Formaldehyde

The effect of the H_2_O_2_ concentration on formaldehyde degradation was investigated in a mixed solution of formaldehyde and glucose. Without the addition of a catalyst, the degradation efficiency of formaldehyde was almost zero. This indicates that in a system without the involvement of the Fe-ZSM-5 catalyst, the degradation of formaldehyde mainly relies on other pathways, such as natural decay, and the efficiency of these pathways is usually negligible. Therefore, we did not consider the independent degradation effect of H_2_O_2_ on formaldehyde in the process of this experiment. When the concentration of H_2_O_2_ increased to 385 mmol L^−1^, it resulted in 74.0% degradation of formaldehyde at the 360 min reaction time (Figure 4a). However, H_2_O_2_ doses of 385 mmol L^−1^ to 476 mmol L^−1^ did not improve the formaldehyde degradation significantly. The positive effect of H_2_O_2_ was attributed to the generation of more hydroxyl radicals. While the concentration of H_2_O_2_ increased to 476 mmol L^−1^, hydroxyl radicals might react inefficiently in the catalytic reaction process under the condition of a constant catalyst dosage [37]. The number of active sites was finite, which led to only a minimal increase in the degradation rate.

The effect of the pH value on formaldehyde degradation was investigated in a mixed solution of formaldehyde and glucose. The catalyst could remove formaldehyde better under acidic than under neutral or alkaline conditions (Figure 4b). This is because the decomposition products of H_2_O_2_ are predominantly •OH under acidic conditions. Alkaline conditions inhibit the generation of •OH, and H_2_O_2_ is easily hydrolyzed to •OOH and H^+^, while the oxidation ability of •OH is much stronger than that of •OOH. When pH = 3, the degradation rate of formaldehyde could reach 75.0% at 360 min contact time. Although the degradation performance was poor in a neutral or alkaline medium, about 65.0%~70.0% degradation rate of formaldehyde could still be reached. The Fe-ZSM-5 heterogeneous Fenton catalyst significantly broadened the pH range of the Fenton reaction and is conducive to applications in other fields.

The effect of reaction temperature on formaldehyde degradation was investigated in a mixed solution of formaldehyde and glucose (Figure 4c). An increase in temperature could accelerate the movement of molecules and the decomposition of H_2_O_2_. With the increase in reaction temperature, the degradation rate of formaldehyde increased and could reach 94.0% at 55 °C at a 360 min reaction time.

Figure 4d shows the effect of catalyst dose on formaldehyde degradation in a mixed solution of glucose and formaldehyde. As expected, the degradation rate of formaldehyde could reach 70.0% with the increase in catalyst dose. The appropriate increase in catalyst dose provided more adsorption sites, which could adsorb more formaldehyde molecules and catalyze the generation of more hydroxyl radicals. However, too much catalyst would promote the occurrence of other side reactions, thereby inhibiting the occurrence of catalytic reactions [36]. An increased Fe^3+^ concentration will inhibit the production of •OH, so an excessive catalyst dose will inhibit the generation of •OH. In a series of Fenton reaction chain reactions, the degradation rate of formaldehyde was relatively reduced when the production rate of Fe^2+^ was lower than the accumulation rate of Fe^3+^.

#### 2.3.2. Selectivity of the Fe-ZSM-5 Catalyst towards Formaldehyde

The selectivity of the catalyst to simulated formaldehyde wastewater was investigated in the mixed solution of formaldehyde and glucose, with both at concentrations of 100 mg L^−1^, 385 mmol L^−1^ H_2_O_2_, 10 g L^−1^ catalyst dose and the reaction temperature 55 °C in a water bath at pH 3 (Figure 5). The 98.2% selectivity to formaldehyde in the heterogeneous Fenton system was better than 86.4% selectivity in the adsorption system. The selectivity coefficients of the catalyst were D_formaldehyde_ = 2.3984, D_glucose_ = 3.9962. The concentration gradient promoted the cycle of adsorption–degradation of formaldehyde, which perpetuated the cycle. This process enhanced the selectivity of the catalyst for formaldehyde by inhibiting the adsorption of glucose.

The stability of heterogeneous Fenton catalysts is of great significance in terms of both economic and environmental factors. This study examined the reusability and stability of Fe-ZSM-5 by measuring the efficiency of formaldehyde degradation and the amount of leaching Fe from Fe-ZSM-5 in multiple cycles (Figure 6a,b). The percentage removal of formaldehyde was more than 90.0% after five cycles, with no significant decline. The average amount of Fe dissolved was 0.048 mg g^−1^ for five continuous cycles, and the average dissolution rate was 0.062%. The iron supported on the catalyst was completely preserved after the reaction, which was probably the reason for its good reuse performance. This further demonstrated that a large amount of iron sludge would not be generated after the reaction. The heterogeneous Fenton catalytic efficiency and good stability of Fe-ZSM-5 indicated that it is appropriate for the removal of formaldehyde in industrial sectors.

#### 2.3.3. Mechanism of the Formaldehyde Degradation

The Fe-ZSM-5/H_2_O_2_ system functioned based on the active radicals generated by the decomposition of H_2_O_2_ to degrade target pollutants. Formaldehyde was degraded by reactive radicals after it was adsorbed by the Fe-ZSM-5, and this reaction occurred on the catalyst surface. Figure 7 shows the effects of adding ascorbic acid, tert-butanol, KI solution, or benzoquinone [18] at a concentration of 10 mmol L^−1^ on the formaldehyde degradation efficiency. The addition of free radical scavengers inhibited formaldehyde degradation. The hydroxyl radical played a dominant role, while peroxide radicals played a limited catalytic role in the reaction process [38]. The degradation of formaldehyde in this reaction system mainly occurred on the surface of this catalyst.

#### 2.3.4. Degradation of the Catalyst on a High Concentration of Formaldehyde Wastewater

The degradation performance of this catalyst on a high concentration of formaldehyde was investigated, at solution pH 3, 385 mmol L^−1^ H_2_O_2_, contact time 360 min and reaction temperature 55 °C in a water bath (Figure 8). The catalyst still had a good degradation performance on the high concentration of formaldehyde-simulated wastewater, and the degradation rate was more than 95.0%. It can be seen that the catalyst not only addressed the defects of the traditional homogeneous Fenton reaction but also had a good degradation performance for a high concentration of formaldehyde-simulated wastewater.

## 3. Materials and Methods

### 3.1. Chemicals

ZSM-5, ammonium acetate, acetylacetone, hydrochloride, iron (III) nitrate nonahydrate (>98%), tert-Butanol, p-Benzoquinone (97%), and sodium hydroxide (>96%) were purchased from Miacklin; Glucose Anhydrose (98%), hydrogen peroxide (30%), 1,10-Phenanthroline, ascorbic acid (>90%) were obtained from Sinopharm Chemical Reagent Co., Ltd. (Shanghai, China); Sodium acetate anhydrous, acetic acid, formaldehyde (37%), sulfuric acid (98%), hydrochloric acid (36%) were supplied by Sinopharm Chemical Reagent Co., Ltd. (Shanghai, China); potassium iodide was obtained from Merck (Shanghai, China).

### 3.2. Preparation of the Fe-ZSM-5 Catalyst

First, 2 g ZSM-5 molecular sieve was added to Fe(NO_3_)_3_ solution (0.8 mol L^−1^) at a liquid–solid ratio of 15:1 to obtain the Fe-ZSM-5 catalyst after 4 h at water bath temperature of 50 °C. The concentration difference of iron ions in Fe(NO_3_)_3_ solution before and after impregnation was determined, and the iron load on the catalyst was calculated.

Then, in order to study the best impregnation concentration of iron ion, the impregnation concentration of iron ion was changed to 0.1 mol/L, 0.2 mol/L, 0.4 mol/L, 0.8 mol/L, and 1.6 mol/L, respectively. The influence of impregnation temperature on the iron loading of the ZSM-5 molecular sieve was studied by changing the impregnation temperature within a gradient range of 25 °C to 95 °C. The ratio of impregnating liquid to solid was changed to 10:1, 15:1, 20:1, 25:1, and 30:1 to investigate the optimal ratio of impregnating liquid to solid. Finally, the concentration difference of iron ions in Fe(NO_3_)_3_ solution before and after impregnation for 1 h, 2 h, 4 h, and 8 h was measured, and the iron load on the catalyst was calculated to obtain the best impregnation time.

### 3.3. Characterization

The microstructure characteristics of the samples were observed by a JSM-7600F thermal field-emission scanning electron microscope (FESEM). The Brunner–Emmett–Teller (BET) surface area and pore volume of the samples were measured by a Micromeritics ASAP-2020 analyzer, while the samples were dried at 110 °C for 6 h. The X-ray diffraction (XRD) pattern of the samples was measured using an Ultima IV assembled multifunctional X-ray diffractometer at a scanning rate of 8°/min and a scanning range of 5° to 80°. The Fourier-transform infrared (FT-IR) spectrum of the samples was measured using a Nicolet iS5 infrared spectrometer to qualitatively analyze the functional groups and chemical bond compositions of materials and to deduce the structures of materials and molecules, and the powder samples were tested after drying under ATR.

### 3.4. Heterogeneous Fenton Catalytic Performance

The Fe-ZSM-5 catalyst was evaluated for the heterogeneous Fenton degradation of formaldehyde at different H_2_O_2_ concentrations, pH, temperature, and catalyst dosages, respectively. The calculated amounts of catalyst and H_2_O_2_ were added to a solution containing equal molar concentrations of glucose and formaldehyde, and the reaction was carried out under constant temperature conditions afterwards. At set intervals, glucose concentration was determined by liquid chromatography. The characteristic absorbance of formaldehyde solution was measured by a UV–vis spectrophotometer at wavelength 414 nm, for which appropriate amounts of acetyl acetone and ammonium salt were added into the separation solution. The selectivity of the catalyst to the target pollutant could be expressed as follows:(1)$$\mathrm{Selectivity}\;\%=\frac{A}{B}\times 100\%$$
where A and B are the concentrations (mg L^−1^) of the target pollutants adsorbed and the total pollutant adsorbed, respectively. The higher the calculated selectivity value, the better the selectivity of the catalyst to the target pollutant. The selectivity coefficient (D) was also used to evaluate the selectivity of catalyst. From Appendix A, it can be observed that the adsorption of glucose and formaldehyde by the Fe-ZSM-5 catalyst was more consistent with the quasi-second-order kinetic equation. Therefore, the strength of the catalyst’s selectivity for formaldehyde is measured by the ratio of quasi-second-order kinetic reaction rate constants. The value of D represents the level of selectivity. The selectivity coefficient can be expressed as follows:(2)$$D=\frac{K_{1}}{K_{2}}$$
where K_1_ and K_2,_ respectively, represent quasi-second-order kinetic reaction rate constants.

### 3.5. Stability and Reusability Measurements

The stability and reusability of this catalyst were measured at a mixture of glucose 100 mg L^−1^ and formaldehyde 100 mg L^−1^, solution pH 3, 10 g L^−1^catalyst dose, 385 mmol L^−1^ H_2_O_2_, contact time 360 min, and reaction temperature 55 °C in a water bath. After each heterogeneous Fenton catalytic cycle, this catalyst was centrifuged from the solution while the eluted Fe and the concentrations of residual formaldehyde in the solution were determined by phenanthroline spectrophotometry [39] and acetyl acetone spectrophotometry [40] respectively. The separated catalyst was washed and dried for 8 h at 110 °C to be used for the next catalytic cycle, with a total of 5 cycles. The dissolution rate of iron was calculated by the formula as follows:(3)$$\mathrm{Iron}\;\mathrm{dissolved}\;\mathrm{rate}\;{\mathrm{mg}\;g}^{-1}=\frac{c_{\mathrm{post}}\times V_{\mathrm{post}}}{W_{\mathrm{catalyst}}}$$
where $W_{\mathrm{catalyst}},$ $c_{\mathrm{post}}$, and $V_{\mathrm{post}}$ represent the mass (g) of granular catalyst added, the concentration (mg L^−1^) of iron, and the residual volume (L) of the solution after the reaction, respectively.

## 4. Conclusions

The Fe-ZSM-5 catalyst was prepared by impregnating the ZSM-5 molecular sieve in ferric chloride solution. Under the conditions of solution pH 3, 385 mmol L^−1^ H_2_O_2_, 10 g L^−1^ catalyst dose, and reaction temperature 55 °C in a water bath, 93.7% of formaldehyde was degraded, and the selectivity of the catalyst for formaldehyde was 98.2% after 360 min. Notably, compared with traditional Fenton technology, the formaldehyde degradation rate of selective heterogeneous Fenton could reach about 75.0% in neutral and alkaline solutions, which indicated that its pH requirements were more relaxed. The degradation rate of formaldehyde could still reach more than 93.0% after five repeated experiments with the catalyst, indicating that this catalyst had good stability. It was speculated that the degradation mechanism of formaldehyde was mainly a surface heterogeneous catalysis mechanism, and the cycle of adsorption–degradation–desorption–re-adsorption–re-degradation–re-desorption was carried out on the catalyst surface. Furthermore, the catalyst prepared in this experiment still had good degradation performance for a high concentration of formaldehyde.

## Figures and Tables

**Figure 1 molecules-29-02911-f001:**
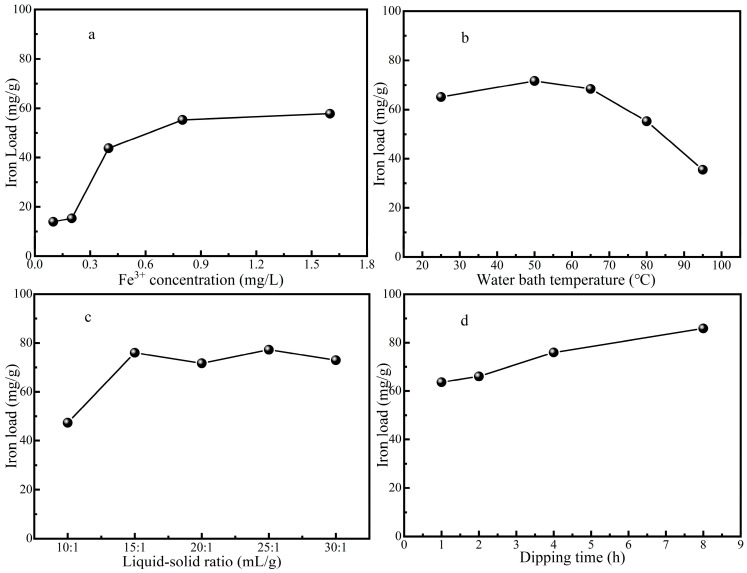
The influence of impregnation concentration (**a**), impregnation temperature (**b**), impregnation liquid–solid ratio (**c**), and impregnation time (**d**) on iron load.

**Figure 2 molecules-29-02911-f002:**
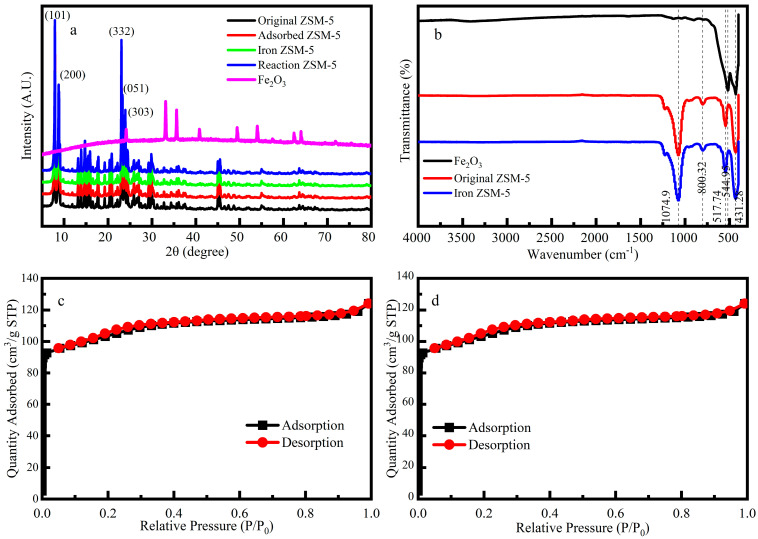
(**a**) XRD patterns of the ZSM-5 zeolite before and after loading iron and Fe-ZSM-5 catalyst after Fenton reaction. (**b**) The FT-IR spectrum of the Fe_2_O_3_ powder the original ZSM-5 zeolite and the Fe-ZSM-5 catalyst. (**c**,**d**) The N_2_ adsorption-desorption isotherm of the ZSM-5 zeolite before and after loading iron. P/P_0_ is the relative pressure (where P_0_ = 1 atm and P is the saturation vapor pressure of N_2_ at 77 K).

**Figure 3 molecules-29-02911-f003:**
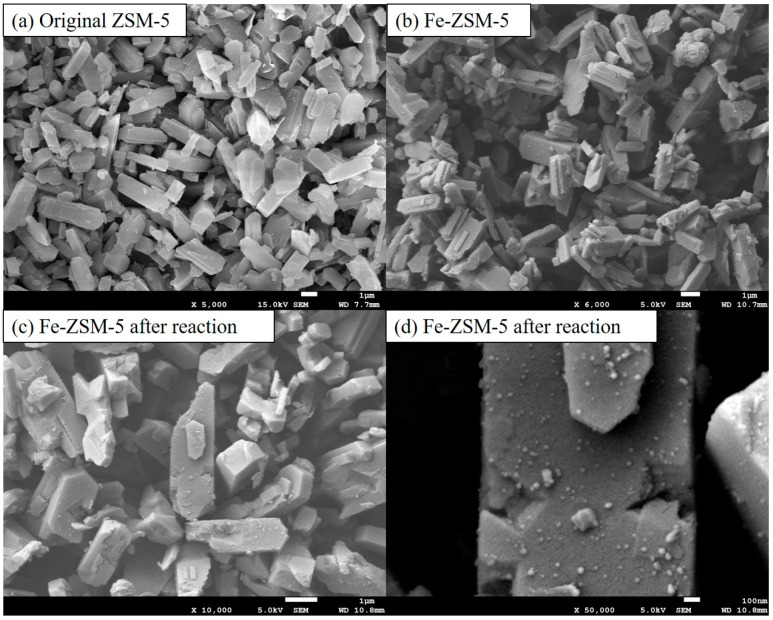
SEM images of (**a**) the original ZSM-5 zeolite, (**b**) the ZSM-5 zeolite after loading iron (**c**) the iron-loaded ZSM-5 molecular sieve catalyst after Fenton reaction. (**d**) shows the iron-loaded ZSM-5 molecular sieve catalyst after Fenton reaction at high magnification (×50,000).

**Figure 4 molecules-29-02911-f004:**
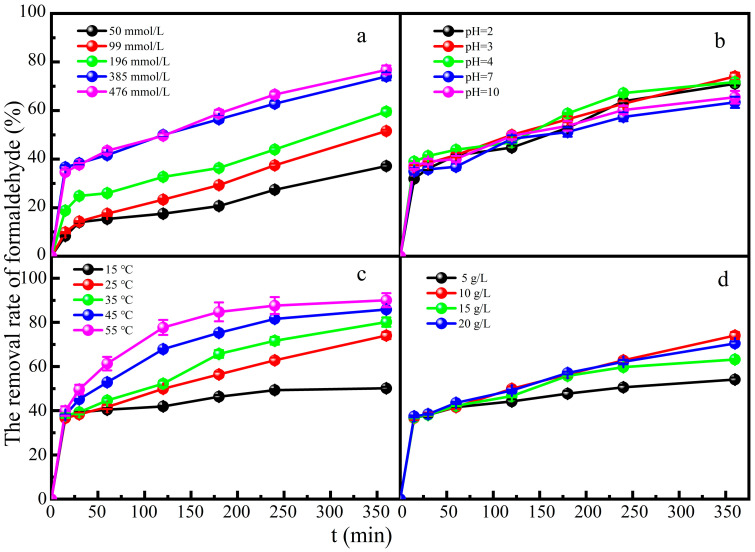
Effect of (**a**) H_2_O_2_ concentration, (**b**) solution pH, (**c**) reaction temperature, and (**d**) photocatalyst dose on the degradation of formaldehyde.

**Figure 5 molecules-29-02911-f005:**
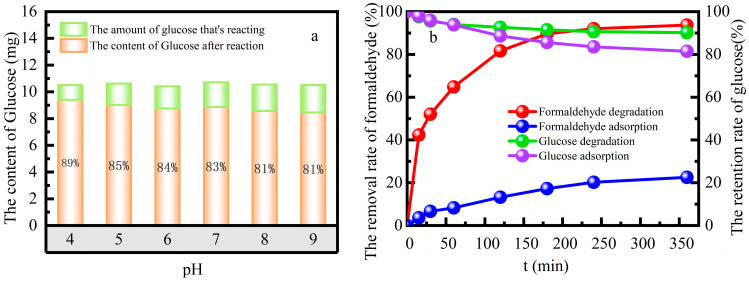
(**a**) Formaldehyde degradation at different pH levels, (**b**) The removal rate of formaldehyde in the adsorption system and the degradation rate in the degradation system. (Initial formaldehyde concentration of 100 mg L^−1^; solution pH 3; H_2_O_2_ concentration of 385 mmol L^−1^; the catalyst dose of 10 g L^−1^; reaction temperature 25 °C; reaction time 0–350 min).

**Figure 6 molecules-29-02911-f006:**
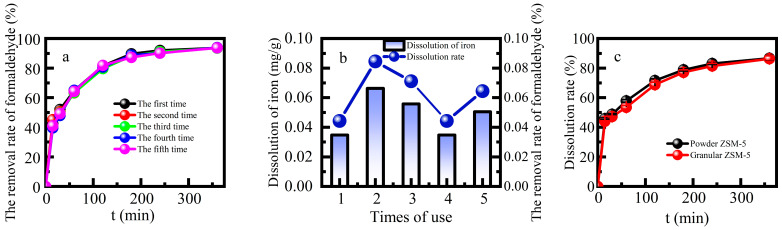
(**a**) Effect of repeated use times on formaldehyde removal rate, (**b**) the iron dissolution amount and (**c**) dissolution rate in the process of repeated use.

**Figure 7 molecules-29-02911-f007:**
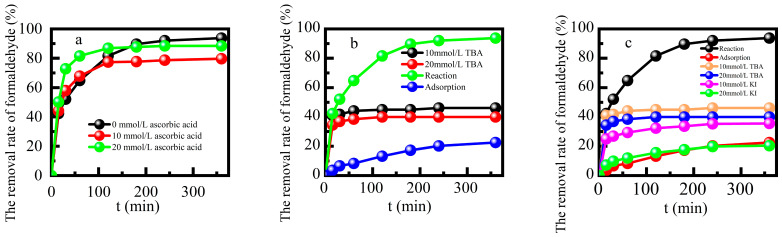
Effects of adding (**a**) ascorbic acid, (**b**) tert-butanol and (**c**) the comparison between KI and p-benzoquinone on formaldehyde removal rate respectively.

**Figure 8 molecules-29-02911-f008:**
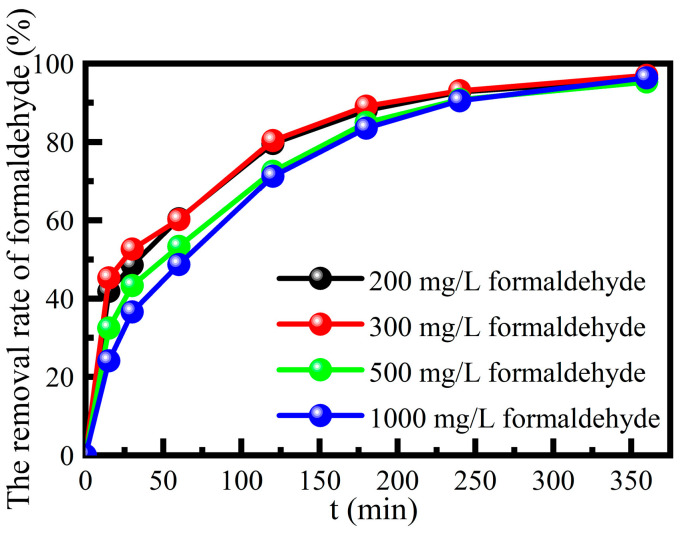
The catalyst degradation of high concentration of formaldehyde-simulated wastewater.

**Table 1 molecules-29-02911-t001:** Textural properties of the original ZSM-5 zeolite and the iron-loaded ZSM-5 molecular sieve catalyst.

Sample	BET Surface Area (m^2^ g^−1^)	Pore Size (nm)	Total Pore Volume (cm^3^ g^−1^)
ZSM-5	654.505	2.329	0.207
Fe-ZSM-5	642.669	2.247	0.205

## Data Availability

The data presented in this study are available on request from the corresponding author.

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
