# Peer review of "Selective Heterogeneous Fenton Degradation of Formaldehyde Using the Fe-ZSM-5 Catalyst"

_molecules, 2024, doi:10.3390/molecules29122911_

Round 1

Reviewer 1 Report

Comments and Suggestions for Authors

Zhou and coworkers reported the synthesis of a novel hetereogeneous Fe-ZSM-5 catalyst and its application in Fenton degradation of formaldehyde. The catalyst is well characterized and showed inpressive reactivity and selectivity. Moreover, the catalyst can be reused at least 5 times. It’s an obviously nice work.

Minor things:

1.     The first sentence In paragraph 2 in the introduction part, the related refs should be recited.

2.     About ZSM-5 in the introduction part, the authors should give the full name.

3.     This work is similar to the work in ref 5, the authors should talk about more about the advantages of this paper.

In summary, this paper can be accepted after minor revision.

Author Response

Thank you for your comments on our manuscript entitled "Selective Heterogeneous Fenton Degradation of Formaldehyde Using the Fe-ZSM-5 Catalyst" (molecules-2985643). Those comments are all valuable and very helpful for revising and improving our paper, as well as the important guiding significance to our researches. We have studied comments carefully and have made correction which we hope meet with approval. Revised portions are marked in red in the manuscript. The main corrections in the paper and the responses to the reviewers' comments are as follows:

  1. The first sentence In paragraph 2 in the introduction part, the related refs should be recited.

Response: Thank you for your suggestion. Adding relevant literatures is very important for writing academic papers. We have added relevant literature (7-10) to support the points in this section.

  1. About ZSM-5 in the introduction part, the authors should give the full name.

Response: Thank you for your suggestion. We apologize for not specifying the full name in the previous version of the article, and we have added the full name of ZSM as “The zeolite molecular sieve” in line 73.

  1. This work is similar to the work in ref 5, the authors should talk about more about the advantages of this paper.

Response: Thank you for your suggestion. Our research has an advantage over the findings reported in literature 5 in that: The Fe-ZSM-5/H2O2 Fenton system can selectively degrade formaldehyde in wastewater thereby enhancing the biodegradability of formaldehyde wastewater. Therefore this system provides the possibility for the efficient and low-energy treatment of formaldehyde wastewater, and can achieve sustainable development by reducing energy consumption and minimizing secondary environmental pollution.

Reviewer 2 Report

Comments and Suggestions for Authors

The manuscript “Selective heterogeneous Fenton degradation of formaldehyde using the Fe-ZSM-5 catalyst,” submitted by Professor Zhou and coworkers, is a valuable contribution with very interesting chemistry. This will certainly help to remove formaldehyde more effectively. The paper is well-written and supported by high-quality experimental data. The authors, however, should add comments on the scalability of the catalyst. Can this catalytic cycle be performed in flow synthesis for potential scale-up? Figure 4 shows the effect of H2O2 concentration on the removal rate of formaldehyde. What if we perform the reaction in the absence of a catalyst? What is the driving force for the mechanism of adsorption-degradation-desorption-re-adsorption-re-degradation-re-desorption? I recommend acceptance subject to satisfactory adaptations of that section as requested.

Author Response

Thank you for your comments on our manuscript entitled "Selective Heterogeneous Fenton Degradation of Formaldehyde Using the Fe-ZSM-5 Catalyst" (molecules-2985643). Those comments are all valuable and very helpful for revising and improving our paper, as well as the important guiding significance to our researches. We have studied comments carefully and have made correction which we hope meet with approval. Revised portions are marked in red in the manuscript. The main corrections in the paper and the responses to the reviewers' comments are as follows:

  1. The authors should add comments on the scalability of the catalyst.

Response: Thank you for your careful review of our manuscript. We sincerely apologize for the lack of introduction to the scalability of this catalyst in the field of catalysis in the previous article, and we have added the relevant content in paragraph 5 in the introduction part. The porous structure, acidic properties, thermal stability, and tunability of ZSM-5 make it widely used in the field of catalysis. Moreover, ZSM-5 can be modified in various ways, such as ion exchange, metal doping, and pore expansion, to enhance its catalytic performance or adapt to specific catalytic processes. In addition, ZSM-5 can effectively catalyze the degradation of organic pollutants in wastewater, promoting green chemistry and sustainable development.

  1. Can this catalytic cycle be performed in flow synthesis for potential scale-up?

Response: Thank you for your careful review of our manuscript. The Fe-ZSM-5 catalyst is a powder-form catalyst commonly used for catalytic oxidation reactions, such as the removal of organic pollutants in wastewater treatment. Although the powder-form catalyst has high catalytic activity, its separation and recovery in practical industrial applications are often challenging, which limits its use in dynamic reactors.

However, our research group has conducted relevant studies: by using granulation technology to convert the powder into granules. This not only facilitates operation and recycle in the reactor but also enhances the mechanical strength and thermal stability of the catalyst. The application of granulated catalysts in dynamic reactors can improve reaction efficiency, reduce costs, and help realize the scaling and industrialization of wastewater treatment.

  1. Figure 4 shows the effect of H2O2 concentration on the removal rate of formaldehyde. What if we perform the reaction in the absence of a catalyst?

Response: Thank you for your careful review of our manuscript. We have conducted relevant experiments, and the results showed that without the addition of a catalyst, the degradation efficiency of formaldehyde was almost zero. This indicates that in a system without the involvement of Fe-ZSM-5 catalyst, the degradation of formaldehyde mainly relies on other pathways, such as natural decay, and the efficiency of these pathways is usually negligible.

However, when we introduced the Fe-ZSM-5 catalyst, the degradation efficiency of formaldehyde was significantly improved. This demonstrates that the catalyst plays a key role in the system, significantly enhancing the activation efficiency of H2O2, and thus achieving effective degradation of formaldehyde. Therefore, we believe that when considering the formaldehyde degradation effect of this system, there is no need to overly focus on the degradation effect of H2O2 on formaldehyde alone.

  1. What is the driving force for the mechanism of adsorption-degradation-desorption-re-adsorption-re-degradation-re-desorption?

Response: Thank you for your careful review of our manuscript. The driving force behind the "adsorption-degradation-desorption-re-adsorption-re-degradation-re-desorption" cycle is primarily due to the selective adsorption of formaldehyde by the Fe-ZSM-5 catalyst. When formaldehyde molecules are adsorbed onto the surface of the catalyst, they are rapidly degraded by hydroxyl radicals (·OH) into harmless or low-toxicity substances. The intermediate products generated in this process then desorb from the surface of the catalyst, releasing adsorption sites. These released adsorption sites, which then re-adsorb formaldehyde molecules from the solution, initiate a new cycle of adsorption and degradation.

Reviewer 3 Report

Comments and Suggestions for Authors

Comments on the Quality of English Language

The authors should check the manuscript carefully to avoid the typos.

Author Response

Thank you for your comments on our manuscript entitled "Selective Heterogeneous Fenton Degradation of Formaldehyde Using the Fe-ZSM-5 Catalyst" (molecules-2985643). Those comments are all valuable and very helpful for revising and improving our paper, as well as the important guiding significance to our researches. We have studied comments carefully and have made correction which we hope meet with approval. Revised portions are marked in red in the manuscript. The main corrections in the paper and the responses to the reviewers' comments are as follows:

  1. The order of Figures is disordered. Please check it carefully.

Response: Thank you for your careful review of our manuscript. We are very sorry for the disorder in the numerical sequence of the figures in the paper, and we have corrected them and marked them in red.

  1. In FT-IR spectra (Fig.2b), the authors attributed the band at 544 cm-1 to the unique chain structure unit of ZSM-5 and Si-O bending vibration of the ZSM-5 molecular sieve. It was very confused. Please explain it.

Response: Thank you for your careful review of our manuscript. We sincerely apologize. This was a typographical error on our part. We have corrected it in line 251. The band detected at 431 cm-1 is related to the Si-O bending vibration in the ZSM-5 molecular sieve.

  1. The SEM element mapping images of ZSM-5, Fe-ZSM-5 before and after reaction should be provided in Fig.3.

Response: Thank you for your suggestion. Upon considering the degradation reaction of formaldehyde (CH2O), we noted that formaldehyde's elemental composition consists of carbon, hydrogen, and oxygen, which are also present in the elemental composition of the catalyst. Additionally, the ZSM-5 and Fe-ZSM-5 catalysts exhibit good stability, so we expect that there will not be significant changes in the elemental distribution before and after the reaction. Therefore, we did not perform an SEM elemental mapping analysis.

  1. There are some typos in this version. Please check it carefully.

Response: Thank you for your careful review of our manuscript. We are very sorry for our typos. We have thoroughly checked the entire text and have made the necessary corrections, marking them in red. For example, we have changed “anticorrosive” to “anti-corrosive” in line 31, “reuse” to “reuses” in line 94 and “rise” to “rose” in line 188.
